# Reducing energy and environmental footprint in agriculture: A study on drone spraying vs. conventional methods

**Mojtaba Safaeinejad, Mahmoud Ghasemi-Nejad-Raeini**[ORCID]*, **Morteza Taki**[ORCID]

Department of Agricultural Machinery and Mechanization Engineering, Faculty of Agricultural Engineering and Rural Development, Agricultural Sciences and Natural Resources University of Khuzestan, Mollasani, Iran

* ghasemi.n.m@asnrukh.ac.ir, ghasemi.n.m@gmail.com

## Abstract

Agricultural practices significantly contribute to resource depletion and greenhouse gas emissions, underscoring the urgent need for environmental sustainability in this sector. This research assesses the energy efficiency and environmental impacts of Unmanned Aerial Vehicle (UAV) technology compared to traditional spraying methods for wheat farms in Lorestan province, Iran. Experiments were conducted randomly with three repetitions, and data were analyzed using Simapro Impact 2002+software, evaluating four primary categories and 15 midpoint indicators. The findings reveal that conventional spraying consumes 2.43 times more energy than drone spraying, with values of 365.26 MJ/ha and 146.84 MJ/ha, respectively. Additionally, the Global Warming Potential (GWP) for pesticide application is 41.284 kg $CO_2$ha$^{-1}$ for conventional methods and 14.485 kg $CO_2$ha$^{-1}$ for drones. Diesel emissions from tractors in traditional spraying represent the most significant environmental burden, while battery production and charging for drones contribute the largest share among various impacts. These results highlight the potential of UAV technology to enhance energy efficiency and reduce environmental harm, promoting sustainable agricultural practices. Nonetheless, battery limitations and the need for training remain challenges, and further studies are required to assess long-term impacts and scalability.

## Introduction

Ensuring an adequate food supply remains a primary concern for researchers despite advancements in various scientific fields. According to the United Nations Food and Agriculture Organization (FAO), the global population is projected to reach 9.7 billion by 2050 and 11.2 billion by 2100 [1]. Agriculture, the world's main source of food [2], faces challenges such as increasing demand, food security, safety concerns, environmental resource protection, and water conservation [3–5]. The uncontrolled use of fertilizers and pesticides due to intensified agricultural activities poses significant

**Data availability statement:** All relevant data are within the manuscript and its Supporting Information files.

**Funding:** The author(s) received no specific funding for this work.

**Competing interests:** The authors have declared that no competing interests exist.

environmental risks. Additionally, limited arable land and a declining number of farmers underscore the need for innovative solutions for sustainable agriculture [6–7]. Addressing these challenges is crucial for a stable and sustainable food supply. The conventional use of chemical toxins raises costs and increases the risk of environmental pollution, potentially causing severe damage due to widespread dispersion [8–9]. Traditional spraying methods apply chemicals uniformly across farms, harming crops, humans, resources, and the environment. Optimal toxin use requires new technologies [10]. Wang et al. (2019) [11] compared the spray deposition, control efficacy on wheat aphids, and working efficiency of UAVs, boom sprayers, and conventional knapsack sprayers. UAVs achieved a control efficacy of 70.9% on wheat aphids, comparable to other sprayers, and demonstrated a working efficiency of 4.11 $hah^{-1}$, 1.7 to 20 times higher than other methods. However, UAVs had a higher deposition variation coefficient (87.2%) compared to boom sprayers (31.2%).

Drones are a promising solution to address these challenges by improving the efficiency of agricultural operations and reducing environmental impact. Their ability to apply pesticides and fertilizers precisely, using low-volume spray methods, enhances sustainability by minimizing resource wastage [12]. The use of drones in agriculture for tasks such as crop monitoring, pest management, and efficient irrigation, enables significant reductions in pesticide usage and energy consumption [13–14]. This contributes directly to environmental sustainability by lowering greenhouse gas emissions, as evidenced by the study conducted by Yuna et al. (2023) [15], which found that drone usage resulted in lower environmental impacts compared to traditional spraying methods. Furthermore, drones offer more efficient land use, especially in smaller farms, where they have been shown to reduce the carbon footprint significantly [16].

While the fundamental nature of agricultural production systems remains constant, maximum productivity can be achieved by adopting the best methods for analyzing these systems [17]. Inefficient resource use in agriculture can lead to severe consequences, and without new technologies and proper management, environmental impacts will escalate [18]. Implementing innovative solutions and effective management strategies is crucial for sustainable practices [19]. Traditional spraying methods require more labor and incur high costs, posing challenges such as labor shortages and increased costs [20]. Agricultural drones offer an effective solution by reducing labor needs and production costs, enhancing productivity, and ensuring consistent application rates [21]. For drone spraying, an automated pre-planned GPS is necessary [22]. Drones replace traditional high-volume spraying with low-volume technology and are used for pest control [23–24]. Remote spraying methods and reduced toxin usage have proven that drones provide better plant protection, making significant progress due to their unique benefits [25].

The integration of Artificial Neural Networks (ANNs) in agriculture, particularly in the context of drone spraying versus conventional methods, plays a crucial role in optimizing processes and reducing environmental footprints [26]. ANNs can analyze vast datasets collected from drone operations, including crop health [27], soil conditions, and weather patterns [28], enabling precise decision-making for pesticide

and fertilizer applications [29]. This precision minimizes chemical wastage and enhances the targeting of specific areas needing treatment, effectively reducing the environmental impact associated with traditional spraying methods [30]. Furthermore, ANNs facilitate the development of advanced algorithms that improve flight path planning for drones, ensuring efficient coverage of fields while avoiding obstacles and minimizing drift [31]. By processing real-time data from sensors onboard drones, ANNs can dynamically adjust spraying parameters such as spray rate and droplet size based on environmental conditions [32]. This adaptability not only enhances the efficacy of pesticide application but also contributes to the sustainability of agricultural practices by lowering chemical runoff into surrounding ecosystems [33].

A study examining the environmental impacts of drone sprayers, tractor-mounted sprayers, and other spraying methods in rice fields across various areas ranging from 5 to 100 hectares in Japan found that drone use had the lowest environmental impact among all methods. In areas smaller than five hectares, the environmental impact of tractor-mounted sprayers was significantly higher, with emissions exceeding $4.142$ kg $CO_2$. The global warming potential of tractor-mounted sprayers was three times greater than that of drones [15]. This highlights the environmental benefits of using drones for agricultural spraying, particularly in reducing carbon emissions and mitigating global warming effects. In a study, Li et al (2024) [34], explores the optimization of spraying missions using multiple UAVs in heterogeneous farmlands with varying pesticide needs. The study demonstrates that an efficiency-first approach significantly reduces pesticide usage and enhances coverage precision. Results indicate improved resource management and minimized environmental impact compared to traditional methods. Another study, evaluates the quality of UAV spraying using a 1D-CNN model and a wireless multi-sensors system [35]. The study found that the model achieved a spraying accuracy of 92.5%, with a coverage uniformity of 87.3%. These results highlight the potential for improved precision and efficiency in agricultural spraying applications.

Recent studies have explored the impact of various UAV flight modes and spray system adjustments on canopy deposition, coverage, and off-target losses in vineyards [36]. This work evaluated twelve combinations of configurations made up of various flight modes, types of nozzles used, and speeds of UAVs. The band spray mode increased the overall canopy deposition from 0.052 to 0.161 µLcm$^{-2}$ (+309%) and reduced the losses on the ground from 0.544 to 0.246 µLcm$^{-2}$ (-54%), considering traditional spray modes as well. On the other hand, a low application rate of a conventional airblast sprayer had better canopy coverage than the best UAV configuration. Another approach proposed a new technique for weed segmentation by leveraging cross-domain transfer learning and achieved an IOU mean of 0.744 for crops while reducing herbicide use by 90% at a spraying resolution of 1.78 × 1.78 cm² [37]. Raj et al. (2024) [38], discussed the integration of UAVs with emerging technologies and reported that UAVs enhanced yield prediction accuracy by 15% and reduced pesticide use by 20%. Cavalaris et al. (2022) [39], compared UAVs and ground sprayers for cotton harvest aid; they found that UAV applications at 2 meters increased canopy deposition by 309%, reduced ground losses by 54%, and enhanced cotton yield. However, Pranaswi et al. (2024) [40], compared UAV with knapsack sprayer for wheat management and recorded the highest growth and yield related to this treatment. Lastly, Meesaragandla et al. (2024) [41], showed that drones spray herbicides 60 times more efficiently and identify weeds compared to traditional methods.

Despite these advancements, the widespread adoption of drone technology in agriculture faces several key challenges. Firstly, the initial costs for purchasing and maintaining drones remain a significant barrier, especially for farmers in developing countries [6]. Secondly, many farmers lack the necessary skills and training to effectively operate drone technology, which highlights the need for comprehensive educational programs to facilitate their adoption [42]. Additionally, concerns over potential drawbacks such as data privacy issues, environmental impacts, and performance limitations under adverse weather conditions further complicate the implementation of drones in agriculture [43]. Addressing these challenges is essential for fostering the successful integration of drone technology into agricultural practices [44].

Based on the above literatures, it is evident that extensive research has been conducted on the functional status of conventional sprayers and the use of new technologies, UAVs drone sprayer, in agriculture. However, given the functional differences, environmental impacts of chemical toxins and varying energy consumption between conventional and

   

drone sprayers, there is a lack of comprehensive studies evaluating and comparing them both technically and environmentally. Conducting thorough technical and environmental evaluations and comparisons of these sprayers is crucial for informed decision-making regarding their application, development, and investment. Therefore, the primary objective of this research is to evaluate and compare the energy consumption and environmental impacts of drone versus conventional sprayers. This will provide a clearer understanding of their respective advantages and disadvantages, aiding in the advancement and adoption of sustainable spraying technologies in agriculture.

## Methodology

The study aimed to evaluate the energy consumption and environmental impacts of using drone spraying technology compared to conventional methods in wheat fields from 10 April 2024–1 June 2024 in northwestern of Iran (Lorestan province). This study consists of three main sections: performance evaluation, energy and environmental evaluation, and effectiveness evaluation of conventional and drone sprayers. All experiments were conducted at three speeds: low, medium, and high, with performance parameters assessed based on these speeds. This research was conducted as part of the first author's PhD dissertation, following the approval of the research proposal by Agricultural Sciences and Natural Resources University of Khuzestan, Iran. The field trials were performed under natural conditions, similar to routine pesticide spraying operations in the region, and no specific permits were required since there was no risk to society or the environment For the energy and environmental evaluation, the commonly used speed in the region was considered as the reference, and the data for these sections were calculated based on the medium speed of the experiments and reported in this paper. Based on previous studies, technical information from sprayer catalogs and experiential data from sprayer operators, it was found that the tractor-mounted sprayer operates at an average speed of approximately 6.5 kmhr$^{-1}$, while the drone operates at an average speed of around 6 ms$^{-1}$ during spraying operations. To evaluate these two sprayers under regional conditions, data for this research were collected and assessed based on these speeds. During the data collection process, each sprayer was individually tested in a one-hectare wheat field with similar and uniform plots and all necessary performance data were measured and recorded. Finally, the data were analyzed using Simapro software. The disposal of drone batteries as a long-term environmental aspect has not been addressed. The experiments for both sprayers were conducted under identical conditions and on the same farms, as shown in **Fig 1**. To ensure uninterrupted spraying operations using drones, it is essential to utilize a portable electricity generator along with three to four spare batteries per drone. This setup enables rapid battery replacement and simultaneous recharging of depleted units, thereby maintaining continuous field performance.

The relevant data for each sprayer are presented in **Table 1**. This table shows the conventional and drone sprayer and all the specifications. All data related to these two sprayers were recorded during the experiments. The input and output energy amounts were calculated by using the energy equivalent of each unit of input or output entities and multiplying this equivalent by the amount consumed or produced during the product's production.

Energy analysis is a valuable tool for assessing agricultural sustainability. Understanding energy inputs and their environmental impacts helps identify pollution sources and supports the development of sustainable farming practices. Evaluating energy balance is thus essential for minimizing environmental damage and promoting resource-efficient agriculture. To analyze energy consumption and evaluate the environmental impacts during the spraying process, all inputs, including machinery (tractor, sprayer, electricity consumption, pesticide, water, and generator) and labor hours, were measured and recorded for two spraying methods (drone-based spraying and traditional tractor-mounted spraying) during the experiments, which were conducted with three replications. The average values obtained were used as the basis for energy calculations and environmental impact assessments. Subsequently, the energy consumption for each input was estimated using energy equivalent coefficients, and the associated environmental impacts were also calculated. **Table 2** provides the energy equations used to calculate the energy for each entity. Finally, the results were used to compare the energy consumption between conventional spraying systems and drone technology. This comparison aimed to highlight the potential benefits of drone technology in terms of energy efficiency and environmental impact.

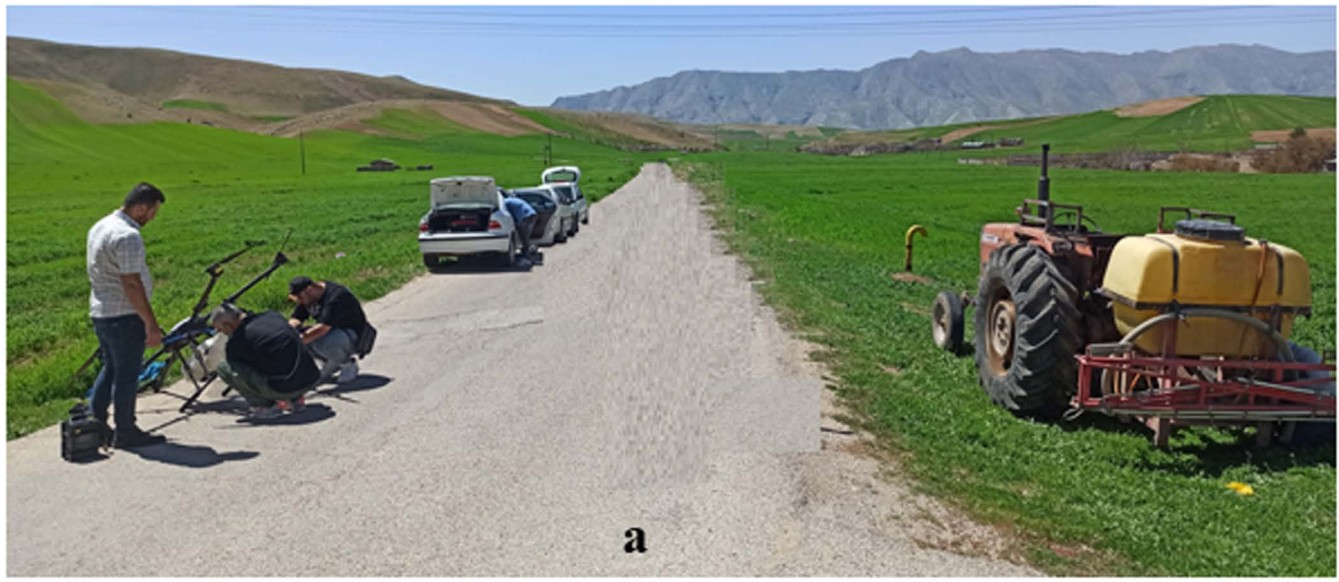

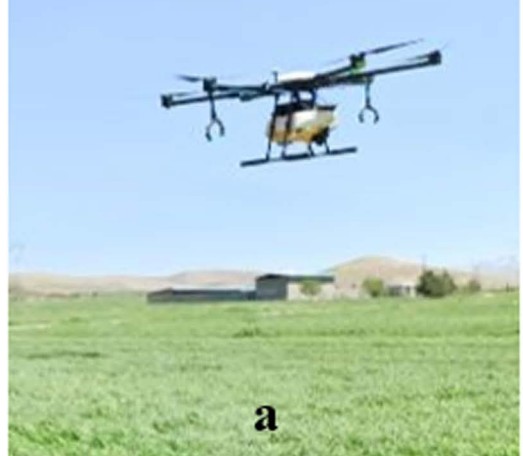

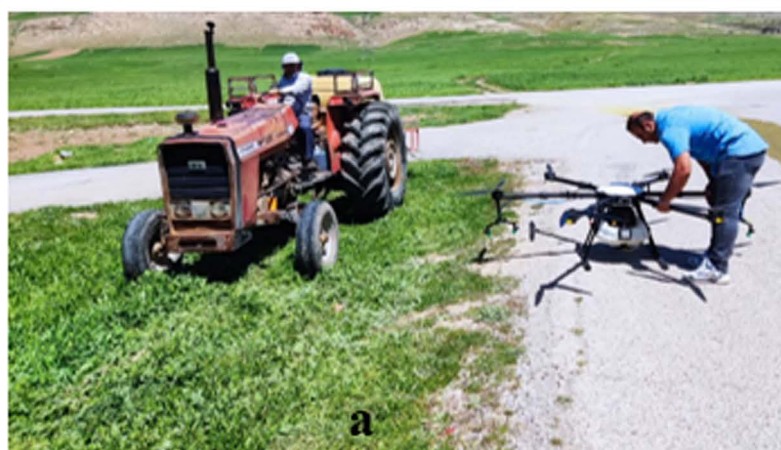

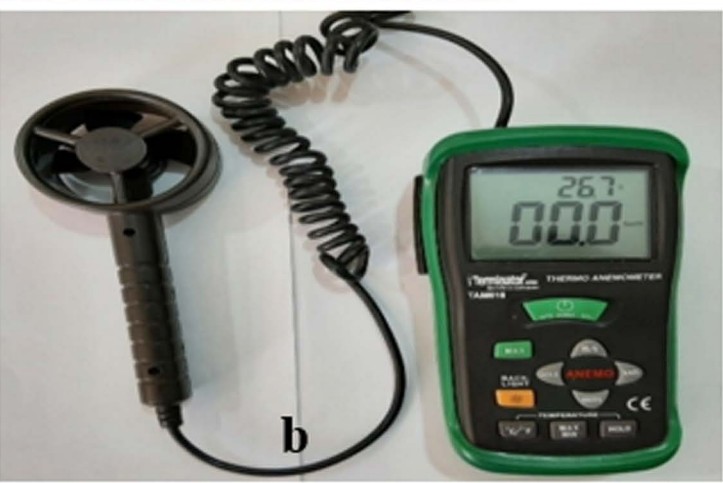

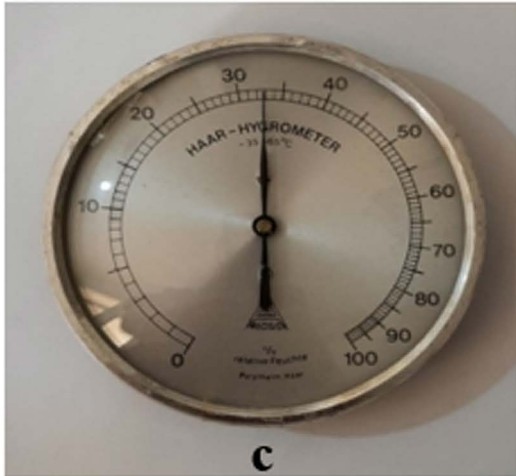

**Fig 1. Environmental measuring tools during field test with sprayers** (a) Wheat field with drone and tractor sprayer, (b) Anemometer and thermometer and (c) Hygrometer.

**Table 1. The conventional and drone sprayer and all the specifications.**

|  | Specifications of the drone |
|---|---|
| Drone Sprayer | The number of motor: 6 |
|  | Flight resistance: 20 Min |
|  | Tank volume:10 L |
|  | Spray width: 5 meter |
|  | Number of cameras: 1 |
|  | Number of LEDs: 2 units for night flight |
|  | Number and type of nozzles: 4 Tee jet |
|  | Maximum spray capacity: $0.5L\ Min^{-1}$ |
|  | Spray Pressure During Test: 1.5 bar<br>Spraying Speed During Test: Three speeds of 21.6, 24.2, and 27 kmh-1 |
|  | Spraying height:3 m |
|  | Spraying height of the drone (SHD):3m |
| **Conventional sprayer** | Specifications of the tractor sprayer |
|  | Tank volume: 400 L |
|  | The material of the tank: Polyethylene |
|  | Pump model: ONTAR 80 |
|  | Boom width: 16 m |
|  | Nozzle type: T jet |
|  | Power train system: PTO |
|  | Pressure relief valve: Manual, three valves |
|  | Pressure range: 1–5 bar |
|  | Spraying Height of the Sprayer (SHS):0.7 m |
|  | Spray Pressure During Test: 1.5 bar<br>Spraying Speed During Test: Three speeds of 4.39, 6, and 8.57 kmh-1 |
|  | Spraying height:0.7 m |

## Evaluate the environmental impact by using Life Cycle Assessment (LCA)

Life cycle assessment (LCA) is an effective environmental management method used to evaluate energy sources, natural resources, and environmental impacts throughout a product's lifecycle [45–46]. LCA involves defining goals, identifying inputs and outputs, and assessing and interpreting environmental impacts, making it valuable for evaluating agricultural products [47–48]. The ISO 14040 and 14044 standards provide guidelines for conducting LCA-based environmental impact assessments, and research typically follows these standards [49–50]. This method considers all sources entering a system and the pollutants released, using various inputs [51–52]. By adhering to these guidelines, LCA offers a comprehensive approach to understanding and mitigating the environmental impacts of agricultural production [53].

According to ISO 14040 and 14044 standards, the main stages of life cycle assessment (LCA) consist of four interconnected phases [49,58]:

1. Goal and Scope Definition: Establishing the purpose and boundaries of the LCA study (**Fig 2**).

2. Inventory Analysis: Identifying and quantifying inputs and outputs within the system.

3. Impact Assessment: Evaluating the potential environmental impacts of the identified inputs and outputs.

4. Interpretation: Analyzing the results to draw conclusions and make informed decisions.

In this study, the first stage involves defining the purpose and scope, including specifying the functional unit, selecting the allocation method, and determining system boundaries. The second stage focuses on inventory analysis, identifying

**Table 2. Equivalents and equations used in calculating the energy of some inputs.**

| Inputs | Unit | Energy equivalent (MJ unit$^{-1}$) | Equations | References |
|---|---|---|---|---|
| Human labor | hr | 1.96 | $E_l = W_l \times E_l$<br>$E_l$: Labor energy (MJha$^{-1}$)<br>$W_l$: Number of workers (nha$^{-1}$)<br>El: Equivalent energy per worker (MJ) | [54] |
| Agricultural machinery | hr | 62.70 | $E_m = H_m \times E_{ma}$<br>$E_m$: Indirect energy of machines (MJha$^{-1}$)<br>$E_{ma}$: Machine energy content (MJhr$^{-1}$)<br>$H_m$: The hours of using machine (Total hours for all operations in one hectare) | [55] |
| Diesel fuel | L | 56.31 | $E_d = Q_d \times E_d$<br>$E_d$: Total fuel consumption (MJha$^{-1}$)<br>$Q_d$:Total fuel consumption (Lha$^{-1}$)<br>$E_d$: Fuel equivalent energy (MJL$^{-1}$) | [56] |
| Pesticide | L | 85 | $E_p = W_p \times E_p$ $E_p$: Consumable pesticide energy (MJha$^{-1}$)<br>$W_p$: Amount of pesticide consumption (Lha$^{-1}$)<br>$E_p$: Energy available in each unit of pesticide (MJL$^{-1}$) | [54] |
| Water for irrigation | M$^3$ | 1.03 | $E_w = W_w \times E_{wi}$ $E_w$: Consumable water energy (MJha$^{-1}$)<br>$W_w$: Amount of water consumption (m$^3$ha$^{-1}$)<br>$E_{wi}$: Energy available in each unit of water (MJm$^{-3}$) | [57] |

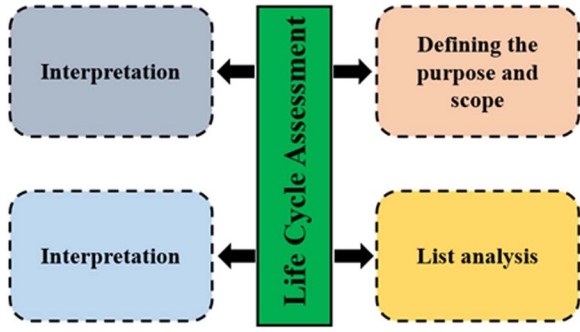

**Fig 2. Life Cycle Assessment stages.**

and quantifying resources used and greenhouse gas emissions [59]. The third stage translates data on resource use and emissions into environmental impacts for each category. The final stage interprets these results to draw meaningful conclusions [60], ensuring a thorough understanding of environmental impacts throughout the product's life cycle [61].

The purpose of LCA in this study was to compare the environmental impacts of drone spraying and conventional (tractor-mounted) spraying. This evaluation followed four steps using Simapro software [62]:

1. Selection and Classification of Impact Categories: Identifying and categorizing relevant environmental impact categories.

2. Characterization of Effects: Quantifying impacts associated with each category.

3.  Normalization: Comparing impacts to a reference value to understand their relative significance.

4.  Weighting: Assigning importance to different impact categories to prioritize them [63].

By following these steps, the study aimed to provide a comprehensive comparison of the environmental impacts of the two spraying methods. The system boundaries for the LCA in the two scenarios (drone-powered spraying and conventional spraying) are illustrated in **Fig 3**, ensuring a thorough comparison of the environmental impacts associated with each method.

This section examines the environmental impacts of chemical toxins, agricultural machinery, and transportation for supplying spraying water by tractors, fuel consumption, and human resources in spraying operations under two scenarios: drone spraying and conventional spraying. Information on the type and amount of each input was collected based on field operations and testing for both methods. Equivalent emissions for each input were obtained using the Ecoinvent database, the table of associations, and other relevant sources. The Ecoinvent database, which contains information on greenhouse gas emissions in the European region, is applicable to the Iranian region and has been used in numerous studies within the country [64–65]. Simapro software was used to evaluate the environmental impact.

## Results and discussion

### Energy analysis

The results of the energy indicator analysis for different spraying methods are presented in **Table 3**. This table illustrates the contribution of each factor to the evaluation process for one hectare, comparing the two scenarios of drone spraying and conventional spraying.

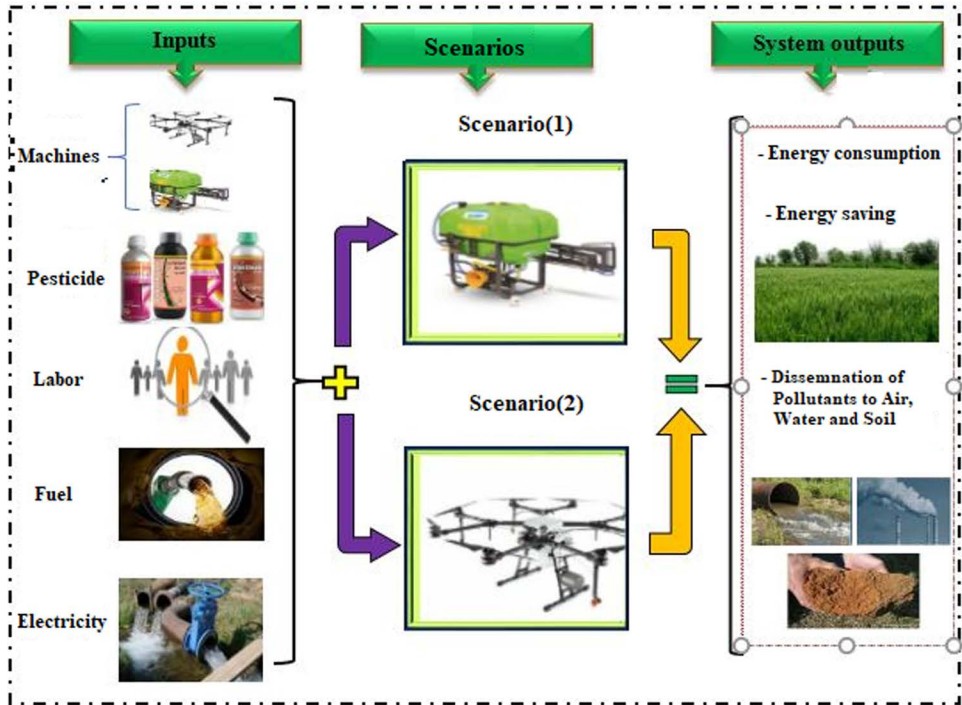

**Fig 3. System boundary in two spraying scenarios using in drones spraying and conventional spraying.**

The results in **Table 3** indicate that energy consumption for drone spraying technology is significantly lower, at 146.84 MJha$^{-1}$, compared to 365.26 MJha$^{-1}$for the tractor-mounted spraying method. Additionally, drone technology enhances agricultural capacity due to time savings, requiring only 0.15 hr ha$^-$ (with a farm capacity of 6.66 ha hr$^{-1}$) compared to 0.5 havhr$^{-1}$ for the tractor-mounted method (with a farm capacity of 2 hahr$^{-1}$). This highlights the efficiency and effectiveness of drone technology in reducing energy consumption and enhancing productivity in agricultural operations. Zarifneshat et al. (2022) [66] evaluated the performance of drone sprayers compared to conventional methods for weed control in wheat fields. The study reported that drone sprayers consumed only 418 kJha-1 of energy, which was significantly lower than that of boom sprayers (2837.8 kJha-1) and Turbo Liner sprayers (4796.2 kJha-1), respectively. Moreover, the drift rate of drone sprayers was 16.76%, compared to 7.66% and 38.6% for boom and Turbo Liner sprayers, respectively. Despite higher initial costs, drone sprayers demonstrated superior energy efficiency and a reduced environmental footprint.

**Fig 4** shows the energy consumption of wheat farm spraying operations using two methods: drone spraying and tractor-mounted spraying. The results indicate that the conventional method consumes approximately 2.48 times more energy than the drone method. When excluding the pesticide, which is common to both spraying methods, this difference increases to 12 times.

This underscores the superior energy efficiency of drone-based spraying over conventional methods. Jebalin et al. (2024) [67] demonstrated that applying 500 g/ha of pre-emergence herbicide pretilachlor with 40 L/ha of spray fluid via drone resulted in greater weed control, higher grain yield, and enhanced energy performance, with an energy use efficiency of 15.11 and energy productivity of 0.42 kg/MJ—significantly outperforming manual spraying techniques. **Table 3** shows the percentage of energy consumed in the spraying operation of one hectare of a wheat farm using the conventional method (tractor-mounted spraying). The results show that fuel consumption in the conventional method accounted for the largest share of total energy consumption, at 225.24 MJha$^{-1}$, representing 62.9% of the total. This high energy consumption is primarily due to the movement and power transfer required by the tractor to operate the sprayer on the farm. Switching to a different spraying method and eliminating the tractor as a power source can significantly reduce energy consumption.

**Table 3** compares the energy consumption of different components involved in drone-based spraying for one hectare of wheat. In this method, the highest share of energy consumption—127.5 MJha$^{-1}$ was attributed to pesticides, accounting for 86.8% of the total input energy, which remained consistent across both spraying methods. This was followed by drone fuel consumption, which amounted to 18.52 MJha$^{-1}$. These results highlight the efficiency of drone spraying, as a major portion of the energy is effectively utilized for the active input (herbicide), thereby reducing the overall energy consumption compared to conventional spraying methods.

**Table 3. Energy input consumption in spraying one hectare of wheat field by drone and conventional methods.**

| Inputs | Unit | Drone spraying | | Conventional spraying | |
|---|---|---|---|---|---|
| | | Value | Energy [MJha$^{-1}$] | Value | Energy [MJha$^{-1}$] |
| Fuel (Tractor) | [L] | 0 | 0 | 4 | 225.24 |
| Fuel (Electricity production generator) | [L] | 0.40 | 18.52 | 0 | 0 |
| Machine (Tractor) | [hr] | 0 | 0 | 0.5 | 8.93 |
| Machine (Spraying) | [hr] | 0 | 0 | 0.5 | 2.3 |
| Drone | [hr] | 0.15 | 0.094 | 0 | 0 |
| Generator | [hr] | 0.50 | 0.125 | 0 | 0 |
| Labor | [hr] | 0.50 | 0.588 | 0.5 | 0.98 |
| Pesticides | [kg] | 1.50 | 127.50 | 1.5 | 127.5 |
| Water | [m³] | 0.01 | 0.01 | 0.3 | 0.309 |
| Total | – | – | 146.84 | – | 365.26 |

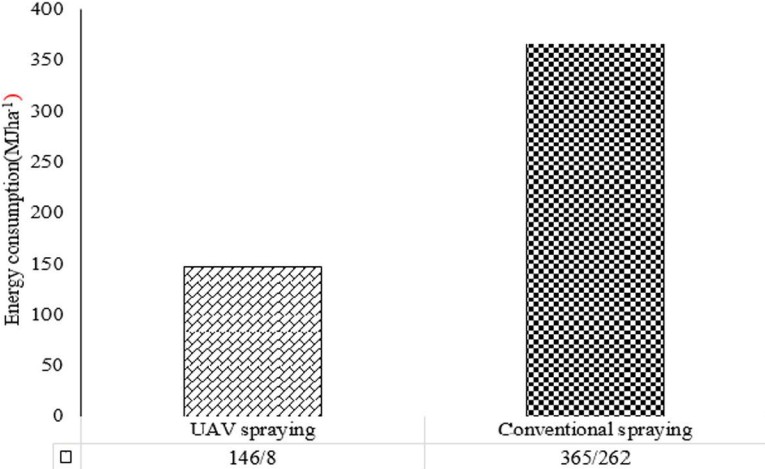

**Fig 4. Compaction of total energy consumption in pesticide spraying of wheat farm by drone and conventional spraying methods.**

**Table 4. Energy indicators in spraying a wheat farm by drone and Conventional method.**

| Index | Unit | Drone spryer | Conventional spryer |
|---|---|---|---|
| Direct energy | [MJ ha$^{-1}$] | 19.11 | 226.22 |
| Indirect energy | [MJ ha$^{-1}$] | 127.73 | 139.04 |
| Renewable energy | [MJ ha$^{-1}$] | 0.59 | 0.98 |
| Non-renewable energy | [MJ ha$^{-1}$] | 146.25 | 363.97 |

**Table 4** presents the energy indicators for spraying one hectare of a wheat farm using two methods: drone spraying and tractor-mounted spraying. In the drone spraying method, renewable energy consumption was 0.59 MJha$^{-1}$, with the total energy consumption estimated at 146.25 MJha$^{-1}$. For the tractor-mounted spraying method, renewable energy consumption was 0.98 MJha$^{-1}$, with the total energy consumption estimated at 363.97 MJha$^{-1}$.

Excluding the pesticide system, which is common to both scenarios, fuel consumption is the primary contributor to renewable energy use in both spraying methods. Reducing or eliminating fuel consumption in spraying could significantly enhance renewable energy savings, promoting more sustainable agricultural practices through drone technology and other innovations. Sahni et al. (2024) [68] examined the efficiency of drone spraying in precision agriculture, finding that it reduced chemical waste by 30% and saved 25% on water usage compared to traditional methods. Additionally, precise targeting and variable rate application improved crop yield by 15%.

## Environmental analysis

The Environmental Impact Assessment results were analyzed across 15 midpoint and four final impact categories. **Fig 5** illustrates the environmental indicators for spraying one hectare of a wheat farm using a tractor-mounted sprayer with 1.5 lit ha$^{-1}$ of the 2, 4-D toxin. This analysis offers a comprehensive overview of the environmental impacts associated with this spraying method, highlighting key areas for potential improvement and mitigation.

The results indicated that the potential for global warming from spraying one hectare of a wheat farm using a tractor-mounted sprayer was 41.28 kg $CO_2$, compared to 14.48 kg $CO_2$ for drone spraying. This significant difference is due to the high fuel consumption of the tractor-mounted method. **Fig 6** shows the environmental impacts across four final categories for tractor-mounted spraying, highlighting the environmental benefits of drone technology in reducing carbon

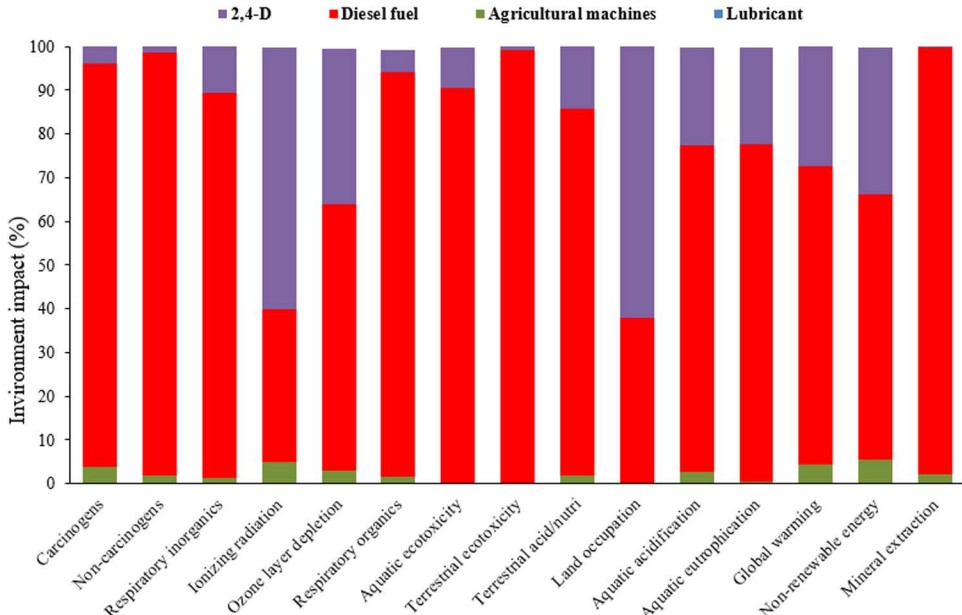

**Fig 5. The results of LCA in 15 midpoint effect categories in spraying of one hectare wheat farm with conventional spraying method.**

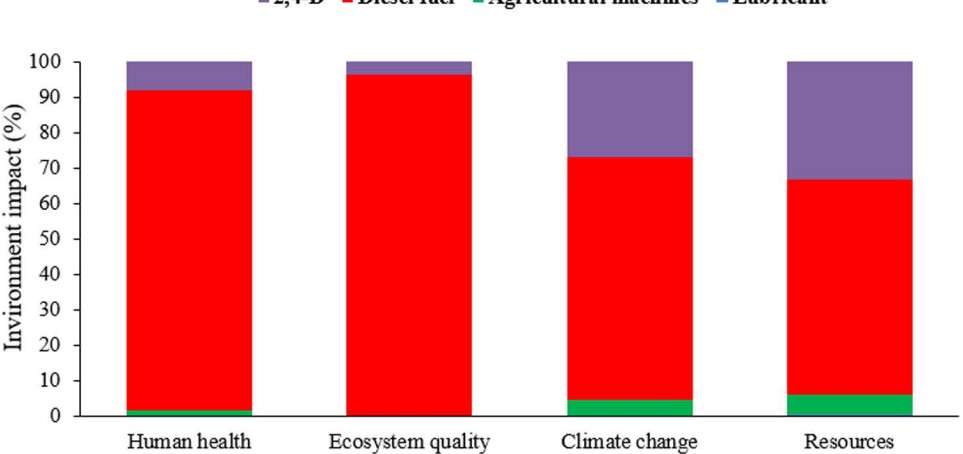

**Fig 6. The results of environmental impact assessment in 4 categories of final effect in spraying of one hectare wheat farm with conventional spraying method.**

emissions. Diesel fuel consumption in tractor-mounted sprayers has the highest environmental impact, with pesticide use ranking second. This method poses a greater risk to human health due to fuel consumption. **Fig 7** shows the environmental impacts in 15 midpoint categories for drone spraying, with battery production and charging accounting for the largest share. Using renewable energy sources, such as solar panels, to charge drone batteries could significantly reduce the environmental impact. This highlights the potential for further improving the sustainability of drone spraying methods. Lan et al. (2024) [69] assessed the environmental and bystander exposure risks from UAV sprayers in golden coconut

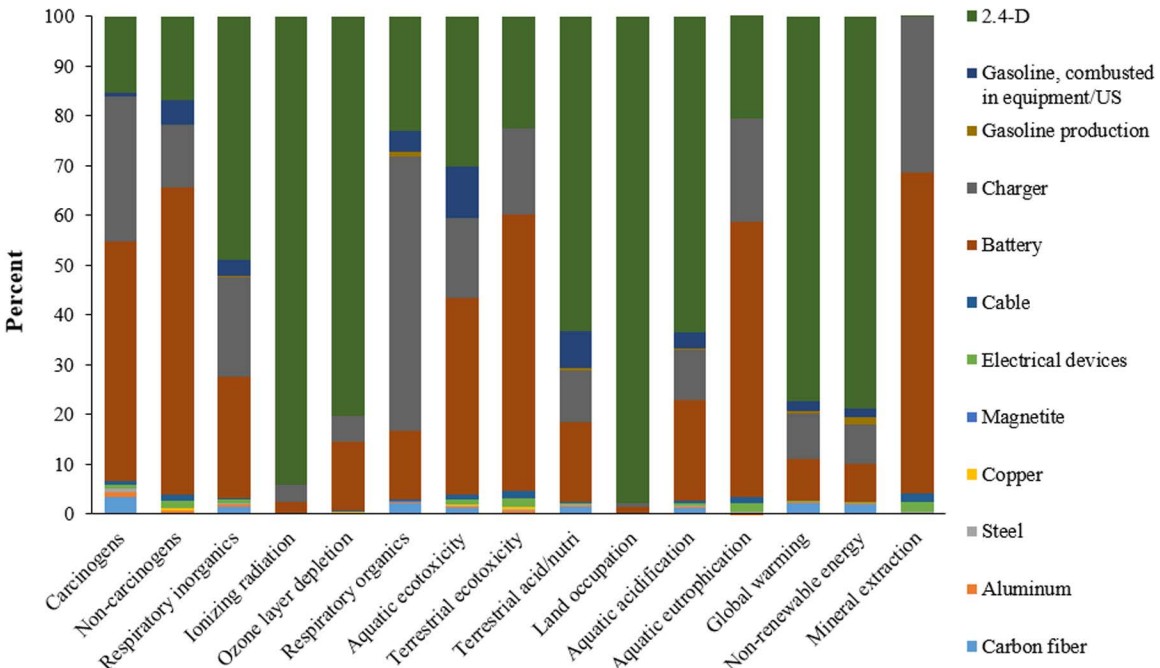

**Fig 7. Environmental impact assessment results in 15 midpoint effects categories in spraying of one hectare wheat farm with drone method.**

plantations, finding that smaller droplet sizes increased spray drift, with buffer zone distances exceeding 30 m and higher cumulative drift percentages. The highest spray drift deposition on bystanders was near chest height.

The environmental impact study demonstrated that using drones for farm spraying results in fewer emissions compared to tractor-mounted sprayers. **Fig 8** shows the environmental impacts of drone use across four final effect categories. Pesticides, battery production, and battery charging were identified as the highest contributors to environmental effects in the drone spraying method. Among these, battery production and charging have a more significant impact on human health. Utilizing renewable energy sources for battery charging could further mitigate these impacts, enhancing the sustainability of drone spraying technology.

**Fig 9** compares the environmental indicators for spraying one hectare of a wheat farm across 15 midpoint effect categories using two methods: tractor-mounted spraying and drone spraying. The results show that, except for the mineral extraction effect, tractor-mounted spraying has a greater environmental impact than drone spraying in all categories, with terrestrial environmental toxicity being the most significant. This highlights the environmental advantages of using drone technology for agricultural spraying, as it generally results in lower environmental impacts compared to traditional tractor-mounted methods. Wang et al (2024) [61] presented a Multi-Objective Teaching-Learning-Based Optimizer (MOT-LBO) for cooperative task allocation between weeding robots and spraying drones. The optimizer aims to minimize the maximum completion time and total residual herbicide. Results showed a 25% reduction in completion time and a 30% decrease in residual herbicide compared to traditional methods.

**Table 5** presents the comparison of environmental indicators across 15 midpoints, evaluating the midpoint effects of using drones versus tractor-mounted sprayers for spraying one hectare of a wheat farm. As shown, the most significant difference between the two methods is in terrestrial environmental toxicity, with tractor-mounted sprayers resulting in 9635 kg TEG soil, compared to just 312 kg TEG soil for drone spraying.

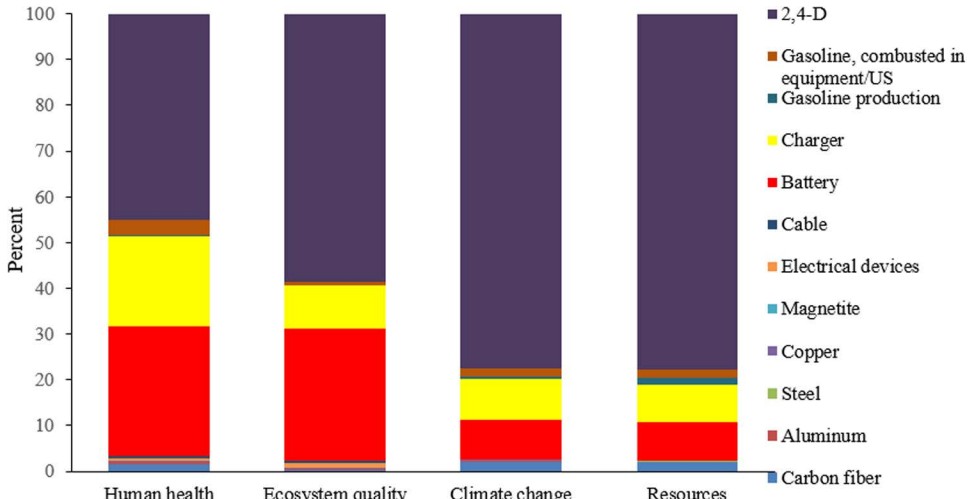

**Fig 8. Environmental impact assessment results in 4 final effect categories in spraying of one hectare wheat farm with drone method.**

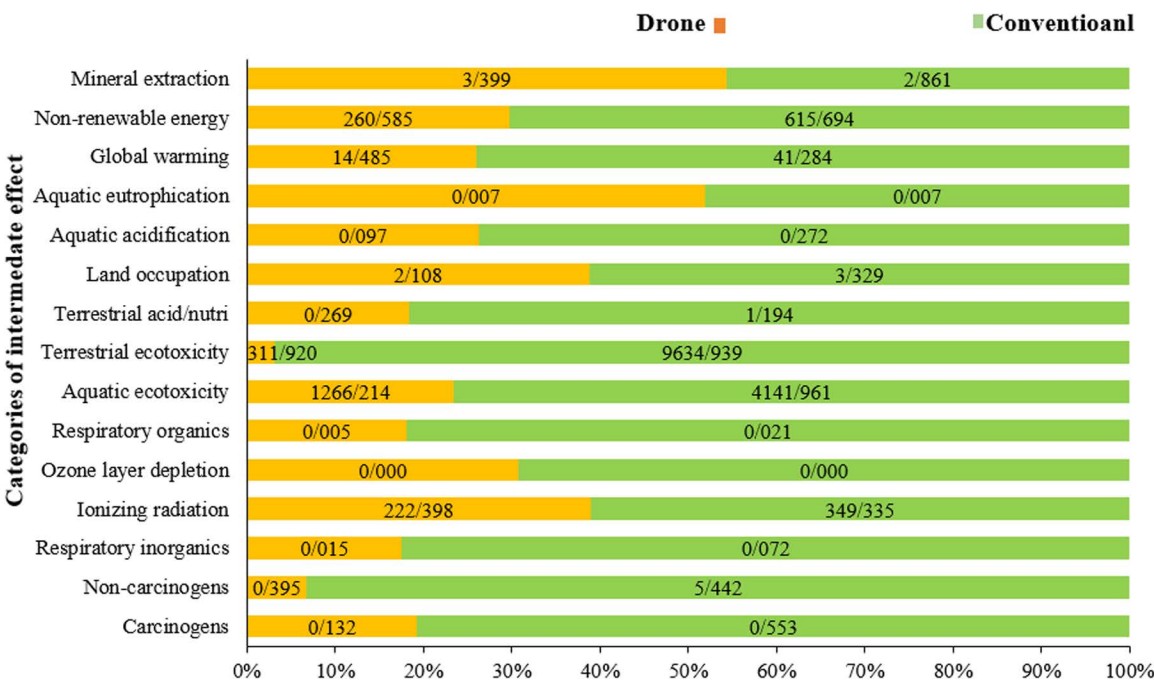

**Fig 9. Results of comparison of environmental indicators in 15 midpoint categories for drone and conventional sprayers.**

**Fig 10** illustrates a comparison of the final effect indicators in the production process of one hectare of a wheat farm, using two methods: drone spraying and tractor-mounted sprayers. The results show that in the final four categories of effects, the environmental impact of drone spraying is significantly lower than that of tractor-mounted sprayers. Among these four categories, the ecosystem quality impact is notably minimal with drone spraying compared to tractor-mounted spraying. Rehman et al (2024) [70] presented an integrated framework combining renewable energy resources, IoT-based

**Table 5. The results of evaluation of environmental effects in 15 midpoint effects categories for drone and conventional method.**

| Effects | Unit | Drone | Conventional |
|---|---|---|---|
| Carcinogens | [kg $C_2H_3Cl$ eq] | 0.132 | 0.553 |
| Non-carcinogens | [kg $C_2H_3Cl$ eq] | 0.395 | 5.442 |
| Respiratory inorganics | [kg PM2.5 eq] | 0.015 | 0.072 |
| Ionizing radiation | [Bq C-14 eq] | 222.398 | 349.335 |
| Ozone layer depletion | [kg CFC-11 eq] | 0.000 | 0.000 |
| Respiratory organics | [kg $C_2H_4$ eq] | 0.005 | 0.021 |
| Aquatic ecotoxicity | [kg TEG water] | 1266.214 | 4141.961 |
| Terrestrial ecotoxicity | [kg TEG soil] | 311.920 | 9634.939 |
| Terrestrial acid/nutri | [kg $SO_2$ eq] | 0.269 | 1.194 |
| Land occupation | [m2org.arable] | 2.108 | 3.329 |
| Aquatic acidification | [kg $SO_2$ eq] | 0.097 | 0.272 |
| Aquatic eutrophication | [kg $PO_4$ P-lim] | 0.007 | 0.007 |
| Global warming | [kg $CO_2$ eq] | 14.485 | 41.284 |
| Non-renewable energy | [MJ primary] | 260.585 | 615.694 |
| Mineral extraction | [MJ surplus] | 3.399 | 2.861 |

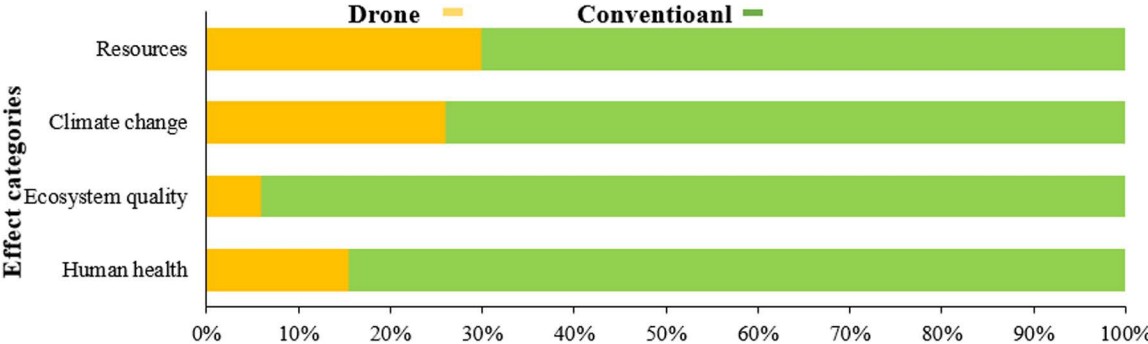

**Fig 10. The results of the comparison of environmental indicators in 4 final effect categories for drone and conventional method.**

energy management, and precision robotics in smart agriculture. The study found that implementing this framework reduced water consumption by 71.8% and energy usage by 30%. Additionally, crop yield increased by 20% due to optimized resource management and precise agricultural practices. These results demonstrate the potential for significant sustainability and efficiency improvements in modern farming.

Fig 11 compare five environmental impact parameters: Terrestrial Ecotoxicity, Non-carcinogens, Global Warming, Non-renewable Energy, and Carcinogens between drone-based and conventional pesticide application methods. These parameters were selected due to their significant differences and critical environmental relevance. Terrestrial Ecotoxicity highlights the direct impact on soil health, with drones significantly reducing soil contamination.

When comparing Carcinogens and Non-carcinogens, both show (Fig 11) substantial reductions in harmful emissions with drone usage compared to the conventional method. However, the reduction is more pronounced in Non-carcinogens. In the drone method, Non-carcinogens emissions are 0.395 kg C2H3Cl eq, which represents a 92.7% decrease compared to 5.442 kg in the conventional method. Conversely, Carcinogens emissions are reduced by 76.1% in the drone method, from 0.553 kg $C_2H_3Cl$ eq to 0.132 kg $C_2H_3Cl$ eq. This comparison illustrates the greater effectiveness of drones in reducing Non-carcinogenic emissions compared to Carcinogenic ones, emphasizing the broader environmental benefits of drones.

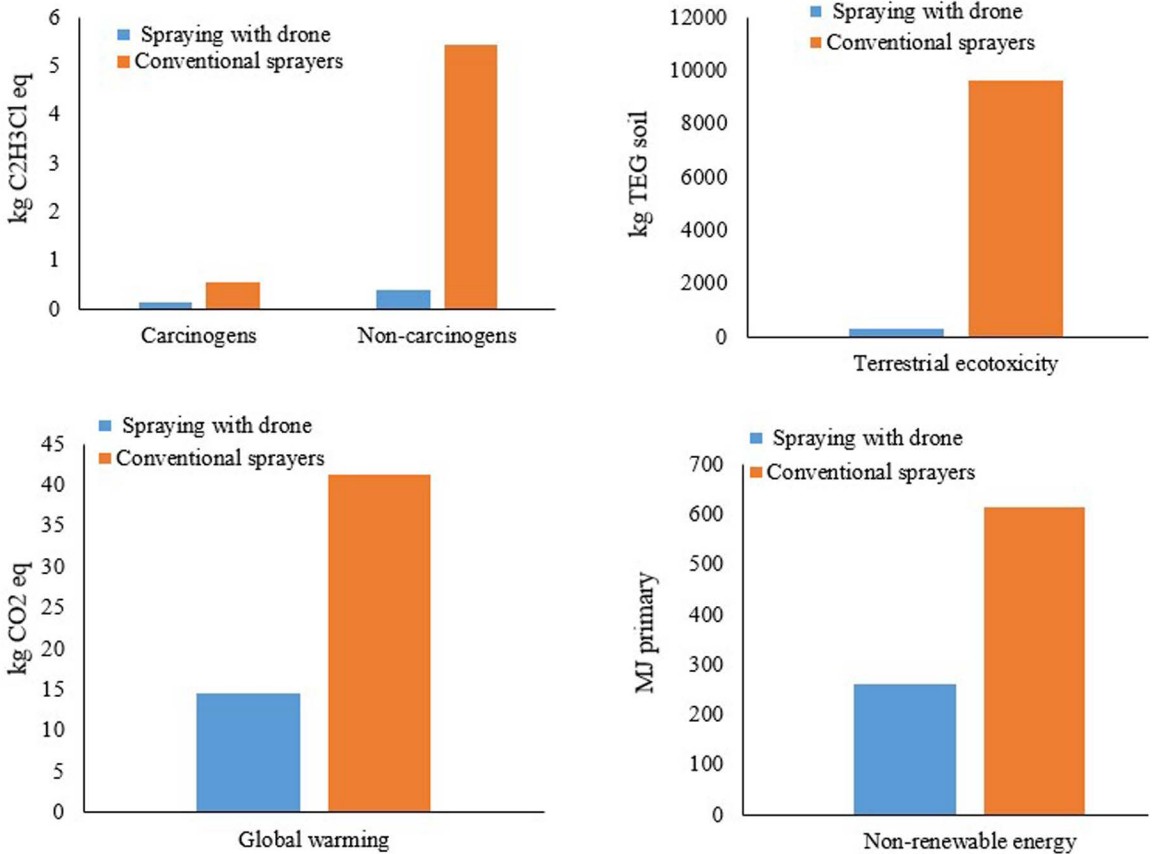

**Fig. 11. Comparison of five environmental impact parameters: terrestrial ecotoxicity, non-carcinogens, global warming, non-renewable energy and carcinogens.**

Global Warming is another significant environmental challenge, where the drone method demonstrates a notable reduction in greenhouse gas emissions. Additionally, Non-renewable Energy consumption is considerably lower with drones, promoting environmental sustainability. These Figs collectively highlight the environmental advantages of drone-based pesticide application over traditional methods.

## Conclusion

The study revealed that energy consumption for spraying one hectare of wheat using a tractor-mounted sprayer (conventional method) was 2.43 times higher than using a drone, with values of 365.26 MJ/ha and 146.84 MJha$^{-1}$, respectively. The potential for global warming from conventional spraying was 41.28 kg $CO_2$, significantly higher than the 14.48 kg $CO_2$ from drone spraying. Diesel fuel consumption in conventional methods had the greatest environmental impact, while battery production and charging were the largest contributors for drones. The operational flexibility of drones allowed for timely interventions, reducing crop damage and improving yield quality. However, practical challenges for drone technology include battery life, operational costs, and the need for specialized training. Long-term environmental impacts of drone usage across various crops should be investigated to understand effects on soil health, biodiversity, and other factors. Additionally, exploring the economic viability and scalability of drones for larger farms would provide valuable insights for widespread adoption. The study demonstrated that drone spraying significantly reduced environmental impact compared to conventional methods, offering actionable insights for practitioners and policymakers in agriculture.

## Supporting information

**S1 Data.**

(XLSX)

## Author contributions

**Data curation:** Mojtaba Safaeinejad.

**Formal analysis:** Mojtaba Safaeinejad.

**Methodology:** Mojtaba Safaeinejad.

**Software:** Morteza Taki.

**Supervision:** Mahmoud Ghasemi-Nejad-Raeini, Morteza Taki.

**Writing – original draft:** Mojtaba Safaeinejad, Mahmoud Ghasemi-Nejad-Raeini, Morteza Taki.

**Writing – review & editing:** Mojtaba Safaeinejad, Mahmoud Ghasemi-Nejad-Raeini, Morteza Taki.

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
