## [Decision Letter · Decision Letter 0]

8 Apr 2025

Dear Dr. Ghasemi nezhad,

Thank you for submitting your manuscript to PLOS ONE. After careful consideration, we feel that it has merit but does not fully meet PLOS ONE’s publication criteria as it currently stands. Therefore, we invite you to submit a revised version of the manuscript that addresses the points raised during the review process.

Overall, a good manuscript for comparing the traditional sprayer with drone sprayers in crop management.

Some aspects needs special attention by the authors for the improved revised version. 

In abstract, findings do not align with conclusion of the study. 

There is more detailed information about the method followed, proper term use and parameters focused. 

There is critical need to check the context of the study with the review and discussion sections. 

We look forward to receiving your revised manuscript.

Kind regards,

Munir Ahmad, PhD

Academic Editor

PLOS ONE

Journal Requirements:

4. Please upload a copy of Figure 14, to which you refer in your text on page 13. If the figure is no longer to be included as part of the submission please remove all reference to it within the text.

Additional Editor Comments:

Overall, a good manuscript for comparing the traditional sprayer with drone sprayers in crop management.

Some aspects needs special attention by the authors for the improved revised version.

In abstract, findings do not align with conclusion of the study.

There is more detailed information about the method followed, proper term use and parameters focused.

There is critical need to check the context of the study with the review and discussion sections.

Discrepancies about some tables and figures, their titles, units, and formatting errors to be addressed.

Reviewers' comments:

Reviewer's Responses to Questions

**Comments to the Author**

1. Is the manuscript technically sound, and do the data support the conclusions?

Reviewer #1: Yes

Reviewer #2: Yes

2. Has the statistical analysis been performed appropriately and rigorously?

Reviewer #1: Yes

Reviewer #2: No

3. Have the authors made all data underlying the findings in their manuscript fully available?

Reviewer #1: Yes

Reviewer #2: No

4. Is the manuscript presented in an intelligible fashion and written in standard English?

Reviewer #1: Yes

Reviewer #2: Yes

Reviewer #1: 1. The first letter in Figure 2 should be capitalized.

2. There is no description of Figure 2 in the Manuscript.

3. The units in Table 3 should be presented in the form of [unit].

4. Please standardize the expressions used throughout the paper (e.g., the use of "Fig." and "Figure").

5. The units in Table 5 should be presented in the form of [unit].

6. In line 351, Figure 14 does not appear in this paper; I believe it should be corrected to Figure 11.

7. Figure 11 and Table 5 convey similar meanings; why are they repeated?

8. In Figure 11, why is the color for “global warming” different? Additionally, it is suggested to modify the vertical axis title to "Name + [unit]", and each subfigure should be labeled (e.g., (a), (b)).

Reviewer #2: Abstract:

1. The finding value do not align with the conclusion. (Row/line - 48)

Methodology:

1. This section does not provide any information on the total number of parameters or how they are measured, which should be clearly explained. (Row/line: 167-169)

2. The term "engine" is not appropriate for this drone. The author should use "motor" instead. (Table no. 01)

3. There is some confusion regarding the drone's height. It should be clarified whether the height was measured from the ground surface or the top of the crop. (Table no. 01)

Results and discussion:

1. The entire study lacks any mention of an electricity generator. The author should either justify its inclusion or remove it. (Table no. 02)

2. The statement and the table present conflicting data, and the author should rectify this discrepancy. (Row/line: 237-238)

3. The energy consumption data is not included in any of the tables. Additionally, the statement references a third spraying method, the turbo liner sprayer, which is not part of the study. The author should correct it. (Row/line: 243-245)

4. The review does not align with the statements and should be placed appropriately. (Row/line: 253-255)

5. The context might be incomplete or incorrect, and the author should either revise or remove it. (Row/line: 263-264)

Conclusion:

1. The data in the statement does not match the script and should be corrected to align with the original. (Row/line: 368-370)

2. The context should clarify and justify how it is measured. (Row/line: 373-374)

**Do you want your identity to be public for this peer review?** For information about this choice, including consent withdrawal, please see our Privacy Policy

Reviewer #1: No

Reviewer #2: **Yes: ** Dr. S. K. Gaadhe

---

## [Author Response · Author response to Decision Letter 1]

11 Apr 2025

Manuscript PONE-D-24-48475

Dear academic editor, Dr. Monika Kundu 

Thank you so much to give me an opportunity to publish my paper in your journal. I received the revise letter with some comments from reviewers and changed the manuscript according to the reviewer’s suggestions and highlighted the changed sentences/words with green paint in manuscript, also I answered the questions from reviewers respectively in this sheet. Thanks a lot for the valuable comments.

Kind regards

M. Ghasemi-Nejad-Raeini

……………………………………………………………………………………………………

Additional requirements Question 1: Ensure the manuscript follows PLOS ONE style requirements, including file naming conventions. Style templates are available at: Main Body Template Title & Authors Template.

Response: Thank you for your valuable feedback on our manuscript. We have carefully reviewed the PLOS ONE style requirements, including file naming conventions, and ensured that the manuscript now fully adheres to these guidelines. We appreciate your attention to detail and guidance.

Additional requirements Question 2: Ensure the manuscript follows PLOS ONE style requirements, including file naming conventions. Style templates are available at: Main Body Template Title & Authors Template.

Response: Thank you for your valuable feedback on our manuscript. We have carefully reviewed the PLOS ONE style requirements, including file naming conventions, and ensured that the manuscript now fully adheres to these guidelines. We appreciate your attention to detail and guidance.

Additional requirements Question 3: In your Methods section, please provide additional information regarding the permits you obtained for the work. Please ensure you have included the full name of the authority that approved the field site access and, if no permits were required, a brief statement explaining why.

Response: Thank you for your valuable feedback on our manuscript. This study is part of the first author’s PhD dissertation, conducted after the research proposal was approved by Agricultural Sciences and Natural Resources University of Khuzestan. The field experiments were carried out under real and natural farming conditions in Iran, exactly similar to routine pesticide spraying practices in the studied region. Given the similarity to regular farming operations, the research posed no risk to the public or the environment. According to national and research regulations, no specific permit was required for conducting such studies. Furthermore, applied research of this kind, which benefits both farmers and the environment, is widely supported and encouraged in Iran.

In addition, the following text has been added to the Methods section: This research was conducted as part of the first author’s PhD dissertation, following the approval of the research proposal by Khuzestan University of Agricultural Sciences and Natural Resources. The field trials were performed under natural conditions, similar to routine pesticide spraying operations in the region, and no specific permits were required since there was no risk to society or the environment.

Furthermore, the Methods section, specifically lines 163 to 166, has been updated to reflect this information.

Additional requirements Question 4: Ensure that the corresponding author has a validated ORCID iD in Editorial Manager. Instructions: ‘Update my Information’ > click 'Fetch/Validate' next to the ORCID field.

Response: Thank you for your observation regarding the corresponding author's ORCID IDs The ORCID ID for the corresponding author, Dr. Mahmoud Ghasemi Nejad Raeini (0000-0001-8945-4049), has been entered and validated in the "Update My Information" section of the Editorial Manager. The information has been updated successfully. We appreciate your attention to this detail.

Additional requirements Question 5: Upload Figure 14 if it is referred to in the text (page 13). If it is no longer part of the study, remove its mention from the manuscript.

Response: Thank you for your thorough and precise review. We sincerely apologize for the typographical error in the manuscript. Upon careful examination, we discovered that on the mentioned page, the reference to Figure 14 was mistakenly included instead of Figure 11. This error has been corrected in the revised manuscript.

The corrected sentence in the manuscript now reads as follows: "When comparing Carcinogens and Non-carcinogens, both show (Fig.11) substantial reductions in harmful emissions with drone usage compared to the conventional method."

We truly appreciate your sharp attention to detail, which has been invaluable in improving the quality of our work. Thank you again for your constructive feedback.

Additional requirements Question 6: Please include captions for your Supporting Information files at the end of your manuscript, and update any in-text citations to match accordingly. Please see our Supporting Information guidelines for more information: http://journals.plos.org/plosone/s/supporting-information.

Response: Thank you for your valuable comment and guidance regarding the Supporting Information files. We have thoroughly reviewed this section and ensured that captions for all Supporting Information files have been included at the end of the manuscript. Additionally, all in-text citations have been updated to match accordingly. We appreciate your careful attention to this matter, which has helped improve the organization and clarity of our manuscript.

Additional requirements Question 7: Please review your reference list to ensure that it is complete and correct. If you have cited papers that have been retracted, please include the rationale for doing so in the manuscript text, or remove these references and replace them with relevant current references. Any changes to the reference list should be mentioned in the rebuttal letter that accompanies your revised manuscript. If you need to cite a retracted article, indicate the article’s retracted status in the References list and also include a citation and full reference for the retraction notice.

Response: Thank you for your thoughtful and detailed comment regarding the reference list. We have carefully reviewed the references to ensure completeness and accuracy. Additionally, any necessary adjustments, including addressing retracted papers and updating citations, have been made with great care. All changes to the reference list have been documented in the rebuttal letter for clarity. We appreciate your guidance in enhancing the precision and reliability of our manuscript.

Additional Editor Comments

Comment 1: Abstract findings do not align with the conclusion. Improve consistency.

Response: The abstract has been carefully reviewed and revised to ensure consistency with the conclusion.

Comment 2: Provide more detail in the methodology, use correct terms and parameters, and ensure they match the context throughout the study.

Response: Additional details have been included in the methodology, particularly regarding certain terms such as "energy." Terms that were previously explained, like "LCA," have been highlighted for clarity in the revised manuscript.

Comment 3: Address inconsistencies in tables and figures: titles, units, formatting, and avoid redundancy.

Response: The formatting of tables and figures has been thoroughly reviewed, with corrections made to titles, units, and any inconsistencies. Additionally, all references to tables and figures in the text have been carefully checked to ensure accuracy and eliminate redundancy. We appreciate the editor's valuable comments and attention to detail, which have been instrumental in improving the overall quality and clarity of the manuscript.

Review Comments to the Author

Reviewer No.1

Question 1: Capitalize the first letter in "Figure 2". Add a description for Figure 2 in the manuscript. Response: Thank you for your precise and constructive feedback. In accordance with your suggestions, "Figure 2" has been updated to "Fig 2" to ensure consistency throughout the manuscript. Additionally, a detailed description for Fig 2 has been added. We greatly appreciate your attention to detail, which has helped enhance the overall quality and coherence of the manuscript.

Question 2: Present units in Table 3 in proper format: [unit].

Response: Thank you for your helpful observation. The units in Table 3 have been reviewed and updated to ensure they are presented in the proper format as per your suggestion. We appreciate your attention to this detail, which has contributed to improving the accuracy and clarity of the manuscript.

Question 3: Standardize terminology throughout the paper (e.g., use either “Fig.” or “Figure” consistently).

Response: Thank you for your detailed observation and thoughtful feedback. The terminology throughout the manuscript has been standardized to ensure consistency, with "Fig." being used uniformly. We appreciate your attention to this matter, which has helped enhance the overall clarity and coherence of the paper.

Question 4: Present units in Table 5 in proper format: [unit].

Response: Thank you for your valuable comment. The units in Table 5 have been thoroughly reviewed and updated to ensure they are presented in the proper format. We appreciate your attention to detail, which has contributed to enhancing the precision and clarity of the manuscript.

Question 5: Line 351: Figure 14 is not present—possibly meant to refer to Figure 11.

Response: Thank you for your attentive and detailed review. We sincerely apologize for the typographical error in the manuscript. Upon review, we confirmed that the reference to Figure 14 was indeed meant to refer to Figure 11. This mistake has been corrected in the revised manuscript. We deeply appreciate your careful observation, which has helped us improve the accuracy and clarity of our work.

Question 6: Figure 11 and Table 5 convey similar meanings; why are they repeated?

Response: Thank you for your insightful comment and detailed observation. While Table 5 provides comprehensive data on 15 items, Figure 11 includes only 5 of these items, specifically those that appeared to show more notable differences and were considered more critical for the evaluation. Including both the table and the figure serves distinct and complementary purposes: the table presents a detailed overview, while the figure visually emphasizes key findings. We believe that the inclusion of both enhances the understanding of the manuscript and aids readers in interpreting the results effectively. Thank you again for your thoughtful review.

Question 7: In Figure 11, why is the color for “global warming” different? Additionally, it is suggested to modify the vertical axis title to "Name + [unit]", and each subfigure should be labeled (e.g., (a), (b)).

Response: Thank you for your thoughtful review and detailed suggestions. The color for "global warming" in Figure 11 has been corrected, ensuring consistency across all elements of the figure. Regarding the vertical axis title and subfigure labels, while we greatly appreciate your recommendation, some of the charts within this figure are paired, making it challenging to uniformly apply these changes without compromising the coherence and uniformity of the figure design. We believe that leaving the figure unchanged is preferable to maintain its clarity and consistency. Additionally, the title of the figure, along with the legend and vertical axis unit, is already explicitly provided, ensuring the necessary information is conveyed clearly.

Once again, we appreciate your valuable feedback and attention to detail, which has been instrumental in improving the quality of our manuscript. Thank you.

Reviewer #2:

Comment 1: The finding value does not align with the conclusion. (Row/line - 48)

Response: Thank you for your precise and thoughtful feedback. Upon reviewing the data and results, we identified a typographical error in the conclusion section, which has now been corrected. The values were originally extracted from Table 3, and the details of these values are clearly presented in the table. The necessary corrections have been made to ensure alignment between the findings and the conclusion. We truly appreciate your careful attention to this matter, which has improved the accuracy of our manuscript.

Comment 2: Methods: Missing total number of parameters and measurement methods (Lines 167–169). Response: Thank you for your attention to this important aspect. While brief explanations regarding this section were previously included in the Methods section, we have now expanded and detailed this part for greater clarity and completeness. Additional explanations have been provided to address the total number of parameters and the measurement methods used. These updates are included in the revised manuscript within lines 184 to 193. We appreciate your thoughtful feedback, which has helped enhance the precision and thoroughness of this section.

Comment 3: Table 1: Replace the term “engine” with “motor” for the drone.

Response: Thank you for your valuable comment. The term "engine" has been replaced with "motor" for the drone in Table 1. We truly appreciate your attention to this detail, which has helped improve the technical accuracy of the manuscript.

Comment 4: There is some confusion regarding the drone's height. It should be clarified whether the height was measured from the ground surface or the top of the crop. (Table no. 01)

Response: Thank you for your insightful comment. The spraying height for the tractor-mounted sprayer was 0.7 meters, while the height for the drone sprayer was set at 3 meters. These heights were established based on the sprayer's manual and typical practice in the spraying region. All spraying operations and experiments were conducted accordingly. The specified heights are clearly presented in Table 1 for reference. We truly appreciate your feedback, which has helped us improve the clarity and accuracy of the manuscript.

Comment 5 its inclusion or remove it. (Table no. 02)

Regarding the Electricity Generator (Table No. 02): We appreciate the reviewer’s insightful comment. The use of an electricity generator is not only relevant to this study but is also essential in practical applications of spraying drones. Each drone battery provides a flight time of approximately 20 minutes and must be recharged frequently to ensure uninterrupted spraying operations. Typically, 3 to 4 batteries are used per drone during fieldwork. Once a battery is depleted, it is immediately replaced, and the discharged unit is recharged on-site using a portable generator. This continuous cycle of battery replacement and recharging is critical to maintaining the operational efficiency of drone spraying. Therefore, the energy consumption associated with the generator is a necessary and realistic component of the energy input calculations for the drone-based spraying system in this study.

Additionally, the Methods section has been updated to include explanations about the use of the electricity generator, which are detailed in lines 174 to 178 of the revised manuscript. Thank you for your valuable feedback, which has contributed to enhancing the clarity and completeness of this study.

Comment 6: The statement and the table present conflicting data, and the author should rectify this discrepancy. (Row/line: 237-238)

Response: Thank you for your careful observation. Upon review, it was confirmed that the table contains the correct data, while the text discrepancy was due to a typographical error. This error has been corrected, and the value of 146.84 has now been accurately reflected throughout the manuscript in all relevant sections. We sincerely appreciate your attention to detail, which has greatly helped improve the accuracy and consistency of the manuscript.

Comment 7: The energy consumption data is not included in any of the tables. Additionally, the statement references a third spraying method, the turbo liner sprayer, which is not part of the study. (Lines: 243–245)

Response: We sincerely appreciate the reviewer’s insightful comment. Energy consumption data for the two spraying methods evaluated in t

---

## [Editor Report · Decision Letter 1]

15 Apr 2025

Reducing energy and environmental footprint in agriculture: A study on drone spraying vs. conventional methods

PONE-D-24-59746R1

Dear Dr. Ghasemi-Nejad-Raeini,

We’re pleased to inform you that your manuscript has been judged scientifically suitable for publication and will be formally accepted for publication once it meets all outstanding technical requirements.

Kind regards,

Munir Ahmad, PhD

Academic Editor

PLOS ONE
---

## [Editor Report · Acceptance letter]

PONE-D-24-59746R1

PLOS ONE

Dear Dr. Ghasemi-Nejad-Raeini,

I'm pleased to inform you that your manuscript has been deemed suitable for publication in PLOS ONE. Congratulations! Your manuscript is now being handed over to our production team.

Kind regards,

on behalf of

Dr. Munir Ahmad

Academic Editor

PLOS ONE